# Switchable bifunctional molecular recognition in water using a pH-responsive *Endo*-functionalized cavity

Xiaoping Wang[1], Mao Quan[1], Huan Yao[1], Xin-Yu Pang[1], Hua Ke[1] & Wei Jiang [1✉]

The construction of water-soluble synthetic hosts with a stimuli-responsive *endo*-functionalized cavity is challenging. These hosts feature a switchable cavity and may bring new properties to the fields of self-assembly, molecular machines, and biomedical sciences. Herein, we report a pair of water-soluble naphthotubes with a pH-responsive *endo*-functionalized cavity. The inward-directing secondary amine group of the hosts can be protonated and deprotonated. Thus, the hosts have different cavity features at the two states and show drastically different binding preference and selectivity in water. We reveal that the binding difference of the two host states is originated from the differences in charge repulsion, hydrogen bonding and the hydrophobic effects. Moreover, the guest binding can be easily switched in a ternary mixture with two guest molecules by adjusting the pH value of the solution. These pH-responsive hosts may be used for the construction of smart self-assembly systems and water-soluble molecular machines.

[1] Shenzhen Grubbs Institute, Guangdong Provincial Key Laboratory of Catalysis, and Department of Chemistry, Southern University of Science and Technology, Xueyuan Blvd 1088, 518055 Shenzhen, China. ✉email: jiangw@sustech.edu.cn

Selective molecular recognition in water is the basis of numerous biological processes[1], such as enzyme catalysis, biological assemblies, transmembrane transportation, signal transduction, and biological machines. Synthetic receptors with the ability of selective molecular recognition in water may provide new tools for chemical biology, environmental sciences, analytical chemistry and biomedical sciences. However, it is difficult for synthetic hosts to achieve selective molecular recognition in water[2–13]. It would require effective use of hydrogen bonding in water, which is however very challenging[14,15]. During the last decades, several biomimetic hosts[16–24] have been developed and they are able to selectively recognize organic molecules in water through shielded hydrogen bonding and the hydrophobic effects. These synthetic hosts show potential in noncovalent bioconjugation[25], chiroptical sensing[26,27], adsorption removal of polar organic pollutants from water[28], and spectroscopic detection of biomarkers[29,30].

Nevertheless, the binding properties of these hosts are not stimuli-responsive. Introducing stimuli-responsiveness would bring new properties in molecular recognition and self-assembly[31–38]. If stimuli-responsive groups are merged into the inward-pointing groups of these hosts, it would provide new tools for switchable noncovalent bioconjugation[39], drug delivery systems[40], water-soluble molecular machines[41–46], and easily recovered host-based adsorption materials[47–49]. To address this, it requires a water-soluble synthetic host with a stimuli-responsive endo-functionalized cavity.

Over the past years, we have reported a series of macrocyclic hosts with an endo-functionalized cavity[24,50]. Of them, water-soluble amide naphthotubes (1a and 1b, Fig. 1a) were shown to be able to selectively recognize a wide scope of organic molecules[51–57] in water by effectively employing shielded hydrogen bonding and the hydrophobic effects. In addition, protonated amine naphthotubes (2a•6H⁺ and 2b•6H⁺, Fig. 1b) are able to bind organic carboxylates in water through buried salt

bridges and the hydrophobic effects[58]. Amine naphthotubes 2a and 2b have the potential to be pH-responsive[59,60], but have to exist in acidic condition to maintain water solubility. Nevertheless, on this basis, new naphthotubes may be designed to realize a stimuli-responsive endo-functionalized cavity.

Herein, we report a pair of naphthotubes with a pH-responsive endo-functionalized cavity which are able to perform switchable bifunctional molecular recognition in water. One amide group and one secondary amine group are incorporated into the naphthotubes as the inward-directing functional groups. In addition, four carboxylate groups are installed as the sidechains to provide water solubility at the pH range in which protonation and deprotonation of the secondary amine group are allowed. Thus, this pair of naphthotubes possesses a pH-responsive endo-functionalized cavity. In the protonated and deprotonated states, the cavity features of the naphthotubes are totally different and thus they show different binding preference and selectivity to guests. The guest binding can be simply switched by adjusting the pH value of the solution.

## Results and discussion

**Design, synthesis and characterization.** Initially, we thought the pH-responsive naphthotubes should be easy to construct. Amine naphthotubes 2a and 2b provide a good basis. In order to maintain the water solubility and to achieve pH responsiveness simultaneously, the water-soluble groups on amine naphthotubes 2a and 2b were changed from the primary ammonium groups to the carboxylate groups, and thus amine naphthotubes 3a and 3b were synthesized. Consequently, the inward-pointing amine groups in 3a and 3b can be protonated and deprotonated by adjusting the pH value of the solution. The p$K_a$ values of the two amine groups in 3b were determined to be 8.7 and 9.9, respectively (Supplementary Figs. 1 and 2). At pH 7.4, both amine groups would be protonated. However, we found protonated 3a•2H⁺ and 3b•2H⁺ undergo serious aggregations in water (Supplementary Fig. 3). The critical aggregation concentration (CAC) for 3a•2H⁺ is even below 0.05 mM, which prevents the study on its molecular recognition property. The aggregation should be caused by the recognition of carboxylate groups in the protonated cavity[58]. Although protonated 3b•2H⁺ has a higher CAC (0.16 mM, Supplementary Fig. 4) at pH 7.4, its binding property is very poor to the common organic guests (for most of the guests, $K_a < 10^3 M^{-1}$, Supplementary Table 1 and Supplementary Figs. 5–21). Therefore, amine naphthotubes 3a and 3b are not appropriate to demonstrate stimuli-responsive molecular recognition. Consequently, new design is needed for realizing a pH-responsive endo-functionalized cavity.

Reducing the number of amine groups in amine naphthotubes 3 may be helpful to solve the aggregation problem. By combining the good binding ability of the amide naphthotubes 1 and the potential in pH-responsiveness of the amine naphthotubes 2, one amide group and one secondary amine group are incorporated into the naphthotubes as the inward-directing functional groups (Fig. 2a). These naphthotubes with a pH-responsive endo-functionalized cavity possess low symmetry and are thus synthetically challenging when compared to the other naphthotubes[24]. The hosts were synthesized by following a step-by-step procedure (see Supplementary Information): the secondary amine group was first constructed by intermolecular imine formation and then reduction; this is followed by intramolecular amide formation through the reaction between primary amine and carboxylate acid.

The syn- and anti-configurational isomers of the naphthotubes (4a and 4b, Fig. 2b) with four ester sidechains were isolated in 8% and 11% yields, respectively, for the two steps. The two

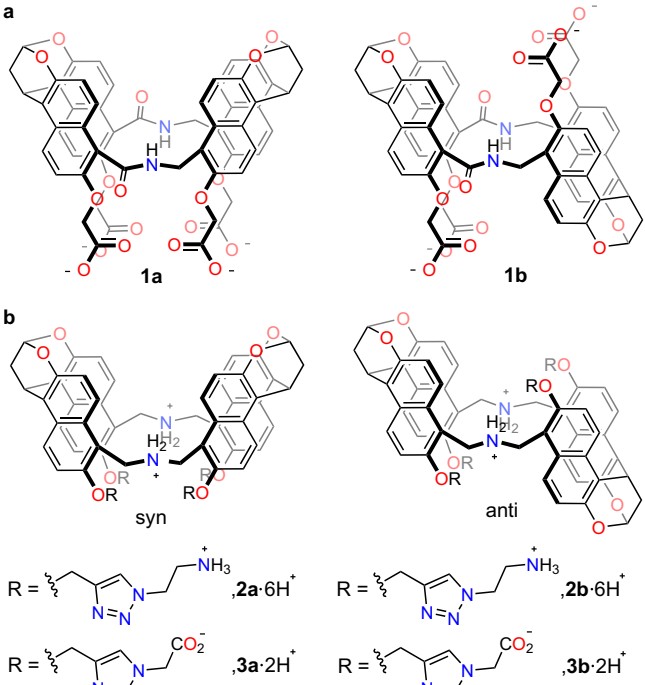

**Fig. 1 Chemical structures of the amide and amine naphthotubes. a** Amide naphthotubes (**1**). **b** Amine naphthotubes (**2** and **3**) with the secondary ammonium groups in the cavities.

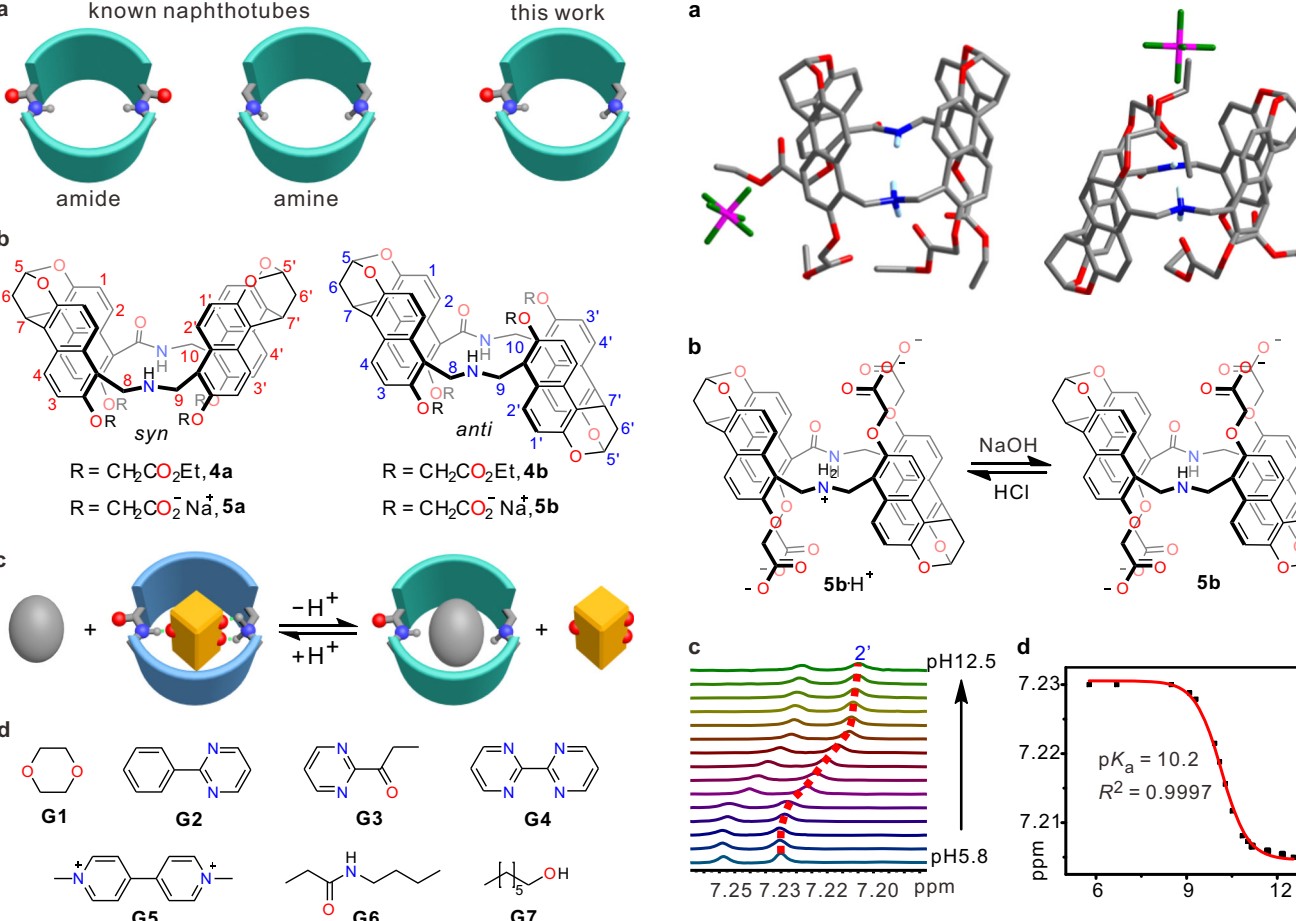

**Fig. 2 Design concept of pH-responsive naphthotubes and chemical structures of the hosts and the guests involved in this research.**
**a** Cartoon representation of the design of the pH-responsive naphthotubes by combining the features of amide and amine naphthotubes. The sidewalls of naphthotubes are colored as celeste; oxygen, red; nitrogen, blue.
**b** Chemical structures of the naphthotubes (**4** and **5**) with the amide and secondary amine groups in the cavities. **c** Cartoon representation of switchable bifunctional molecular recognition in water using the pH-responsive naphthotubes. The sidewalls of protonated naphthotubes are colored as blue. **d** Chemical structures of the guests involved in this research. The counter anion of **G5** is Cl⁻.

**Fig. 3 X-ray single crystal structures of 4 and p$K_a$ value determination of 5b. a** X-ray single crystal structures of **4a** and **4b**. Solvent molecules and most of the hydrogen atoms are omitted for viewing clarity. Color codes: carbon, gray; oxygen, red; nitrogen, blue. **b** Interconversion between the protonated and deprotonated states of **5b**. **c** Partial ¹H NMR spectra of **5b** upon changing the pH values. **d** Titration curve tracking the change in the chemical shift of **5b** with changing the pH values, which was fit to the Henderson-Hasselbalch equation to determine p$K_a$ value and Hill coefficient.

configurational isomers were assigned by 2D NMR spectra (Supplementary Figs. 22–25) and further confirmed by X-ray single crystallography. As shown in Fig. 3a, the secondary amine was protonated; the protons of the amide and ammonium groups were pointing toward the cavity and were thus shielded. The protonation of the amine group may be due to their high p$K_a$ (dibenzylammonium, p$K_a$ = 8.52 in water)[61], and the counter anion PF₆⁻ should originate from the coupling agent PyBOP. After hydrolyzing the ester groups, water-soluble naphthotubes **5a** and **5b** were obtained. They have been further characterized in details by ¹H NMR, 2D NMR spectroscopy, and mass spectrometry (Supplementary Figs. 26–30).

The protonation and deprotonation states of **5a** and **5b** have been carefully studied in water (Fig. 3b). The p$K_a$ values of **5a** and **5b** were determined to be 10.1 and 10.2, respectively, in water by ¹H NMR titrations (Fig. 3c and Supplementary Figs. 31 and 32) through fitting to the Henderson–Hasselbalch equation. The Hill coefficients for these experiments are close to 1.0 (for **5a**, 1.17; for **5b**, 1.24; Supplementary Table 2), supporting a two-state model.

These values were further confirmed by fluorescence titrations (Supplementary Figs. 33 and 34). From the titration curves (Fig. 3d and Supplementary Figs. 31–34), it is clearly seen that the secondary amine group would be fully protonated at pH 7.4 and fully deprotonated at pH 12. Therefore, these two pH values were selected for the following guest-binding studies. These two states may show different binding behaviors (Fig. 2c). Moreover, the CAC of protonated **5a·H⁺** and **5b·H⁺** are more than 0.8 mM at pH 7.4 (Supplementary Fig. 35). All the titration experiments for the determination of association constants were performed at the host concentration lower than 0.5 mM to avoid the interference from aggregation.

**Molecular recognition in water.** The host–guest binding properties were studied in phosphate buffer at pH 7.4 and 12. At both the pH values, the carboxylate groups (p$K_a$ = ~4.8)[61] of the naphthotubes exist in the deprotonated state, thus providing water solubility for the hosts.

**5a** and **5b** have the similar cavity sizes as those of the amide naphthotubes. Therefore, typical guests for the amide naphthotubes, for example 1,4-dioxane (**G1**)[51], aliphatic alcohols (**G7**)[62,63] and phenyl pyrimidine (**G2**)[55] and their analogs (**G3**,

**G4**, and **G6**, Fig. 2d), were selected to study the binding behavior of **5a** and **5b**. In addition, methyl viologen (**G5**) was chosen as a guest as well because the cavities of these naphthotubes are electron-rich in the deprotonated state. These guests maintain in a neutral charge state at the used pH values because their $pK_a$ values are smaller than 2 (Supplementary Table 3).

At pH 7.4, naphthotubes **5a** and **5b** were protonated and thus their cavities are positively charged. Therefore, they may not bind the positively charged guest **G5** but may prefer to bind the guests with multiple hydrogen-bonding acceptors such as **G2–G4**. Indeed, when mixing the hosts and the guests in a 1:1 ratio at pD 7.4, $^1$H NMR signals of both hosts and guests undergo significant changes (Supplementary Figs. 36–49) when compared to those of free guests and hosts. This is true for most of the guests but not for **G5**. This suggests that the protonated states of **5a•**H$^+$ and **5b•**H$^+$ are good hosts for the neutral guests. At pH 12, naphthotubes **5a** and **5b** exist in a non-protonated state. Their cavities are neutral and may prefer to bind hydrophobic guests and even positively charged guests such as **G5**. Except for **G4**, significant changes in the $^1$H NMR spectra were observed (Supplementary Figs. 36–49) when mixing the hosts and the guests in solution at pD 12.

The binding affinities at the two pH values were further quantitatively studied. A 1:1 binding stoichiometry was assumed for all these host–guest pairs, which is supported by Job plots (Supplementary Figs. 50–53) and the knowledge on similar naphthotubes[24]. The association constants of **5a** and **5b** to these typical guests were thus determined at pH 7.4 and 12 by $^1$H NMR titrations (Supplementary Figs. 54–73) and the data are listed in Table 1.

In general, the *anti*-configurational isomer **5b** is a better binder to these guests than the *syn*-configurational isomer **5a** at both pH values. This is similar as the amide naphthotubes[24]. As shown in Table 1, the binding affinities of **5a** and **5b** to the same guests are drastically different at pH 7.4 and 12. The association constants of **G1–G4** are higher at pH 7.4 than those at pH 12. In contrast, **G5–G7** have higher binding affinities at pH 12 than at pH 7.4. The ratios of the association constants at the two pH values for the same host-guest pair were also calculated and listed in Table 1. For the host–guest pairs of **5b** with **G4** and **G5**, the

association constants differ as large as 189 and 295 folds, respectively, at the two pH values. This clearly indicates that it is possible to drastically change the binding affinity of naphthotubes **5a** and **5b** to the same guests by simply adjusting the pH values and thus the protonation state of the host cavities. That is, the complexation of a guest by these hosts can be switched on and off by simply changing the pH value.

As mentioned above, naphthotubes **5a** and **5b** prefer to bind to guests **G1–G4** with multiple hydrogen-bonding acceptors at pH 7.4. The association constants to **G5–G7** are one to three orders of magnitude lower. In particular, no obvious binding was detected between **5b** and **G5**. At pH 12, the association constants of naphthotubes **5a** and **5b** to these guests are at the range of $10–10^4$ M$^{-1}$. **G2** and **G7** are preferred over other guests. However, the binding affinity of **G2** is lower at pH 12 than at pH 7.4, and the binding affinity of **G7** is higher at pH 12 than at pH 7.4. This interesting binding behavior would lay the basis for switching the complexed guests in the cavities of the naphthotubes through adjusting the pH values.

**Binding mechanism**. In order to reveal the thermodynamic binding behavior of these pH-responsive naphthotubes, the association constants and thermodynamic parameters of **5a** and **5b** to **G1–G3** were determined by ITC titrations (Supplementary Figs. 74–87) at both pH 7.4 and 12. These three guests possess good water solubility and show large enough binding affinity for ITC titrations. The corresponding data are listed in Table 2.

The thermodynamic data may provide insights into the binding mechanism. As shown in Table 2, the bindings of all the six host–guest pairs are mainly enthalpy-driven with a favorable or unfavorable entropic contribution. For all the three guests, the association constants are higher at pH 7.4 than those at pH 12. The increased binding affinity at pH 7.4 is mainly originated from enthalpic gain. For guest **G3**, the entropic contribution are similar at both pH values for both the hosts, but their enthalpy differ significantly (for **5a**, $\Delta\Delta H^\circ = -9$ kJ mol$^{-1}$; for **5b**, $\Delta\Delta H^\circ = -9$ kJ mol$^{-1}$). This large difference in the enthalpy at two different pH values should originate from the different noncovalent interactions shielded in the hydrophobic cavity.

Similar as the amide naphthotubes[51–54], shielded hydrogen bonding and the hydrophobic effects through releasing the cavity waters[64,65] or occupying the dry cavity[66] should be the main driving forces for the binding. Direct noncovalent interactions between hosts and guests can be analyzed using X-ray single crystal structures. However, single crystals of the host–guest complexes could not be obtained from water even after many trials. This may be attributed to the high water solubility of **5a** and **5b**. Fortunately, single crystal for the complex of **4b** and **G4** in CH$_2$Cl$_2$, suitable for X-ray crystallography, was obtained. The crystal structure was solved and shown in Fig. 4. Clearly, **G4** is fully encapsulated in the cavity of **4b•**H$^+$. In addition, four hydrogen bonds were detected between the nitrogen atoms of the guest and the amide proton and the ammonium protons of the host. Noteworthily, the hydrogen bonds with the ammonium protons are charge-assisted. These hydrogen bonds should exist even in water because they are shielded in a hydrophobic microenvironment in the cavity[67]. The binding mode and the binding driving force should be still valid in water where the hydrophobic effect also plays an important role. Nevertheless, desolvation of the protonated cavities of the naphthotubes is likely enthalpically more difficult, as indicated by the lower binding affinities to the hydrophobic guests (**G5–G7**) when compared to neutral naphthotubes. Therefore, this explains the reason why the naphthotubes prefer to bind the guests with multiple hydrogen-bonding acceptors at pH 7.4.

**Table 1 Association constants ($K_a$, M$^{-1}$) of naphthotubes 5a and 5b with G1-G7 in phosphate buffer (50 mM, pH = 7.4 or 12, H$_2$O/D$_2$O = 9/1) at 298 K as determined by $^1$H NMR titrations[a].**

| Guest | Host | $K_a$ (pH 7.4) | $K_a$ (pH 12) | Ratio[b] |
|-------|------|----------------|---------------|----------|
| G1 | 5a | $(4.8 \pm 0.6) \times 10^3$ | $(1.6 \pm 0.2) \times 10^2$ | 30/1 |
|    | 5b | $(3.8 \pm 0.7) \times 10^3$ | $(4.9 \pm 0.7) \times 10^2$ | 8/1 |
| G2 | 5a | $(8.4 \pm 0.1) \times 10^4$ [c] | $(2.4 \pm 0.4) \times 10^3$ [c] | 35/1 |
|    | 5b | $(3.7 \pm 0.1) \times 10^5$ [c] | $(1.5 \pm 0.2) \times 10^4$ [c] | 25/1 |
| G3 | 5a | $(1.9 \pm 0.3) \times 10^4$ | $(6 \pm 1) \times 10^2$ | 32/1 |
|    | 5b | $(1.1 \pm 0.1) \times 10^5$ | $(1.8 \pm 0.1) \times 10^3$ | 61/1 |
| G4 | 5a | $(8.6 \pm 0.2) \times 10^3$ [c] | $(5 \pm 1) \times 10^1$ | 172/1 |
|    | 5b | $(1.7 \pm 0.2) \times 10^4$ [c] | $(9 \pm 2) \times 10^1$ | 189/1 |
| G5 | 5a | _[d] | _[d] | _[d] |
|    | 5b | $(2.0 \pm 0.4) \times 10^1$ | $(5.9 \pm 0.9) \times 10^3$ | 1/295 |
| G6 | 5a | $(7 \pm 1) \times 10^1$ | $(7.0 \pm 0.6) \times 10^2$ | 1/10 |
|    | 5b | $(1.0 \pm 0.1) \times 10^2$ | $(1.6 \pm 0.1) \times 10^3$ | 1/16 |
| G7 | 5a | $(1.0 \pm 0.1) \times 10^3$ | $(5.6 \pm 0.7) \times 10^3$ | 1/6 |
|    | 5b | $(2.0 \pm 0.4) \times 10^3$ | $(4.0 \pm 0.4) \times 10^4$ | 1/20 |

[a]The data were averaged from three independent titrations.
[b]Ratio = $K_a$ (pH 7.4)/$K_a$ (pH 12).
[c]These association constants were determined by ITC titrations.
[d]The association constants could not be obtained due to severe aggregation in the host–guest mixture.

**Table 2 Association constants ($K_a$, M$^{-1}$) and thermodynamic parameters ($\Delta H^o$ and $-T\Delta S^o$, kJ mol$^{-1}$) of naphthotubes 5a and 5b with G1–G3 in phosphate buffer (50 mM, pH = 7.4 or 12, H$_2$O) at 298 K as determined by ITC titrations[a].**

| Guest | Host | pH 7.4 | | | pH 12 | | |
|---|---|---|---|---|---|---|---|
| | | $K_a$ | $\Delta H^o$ | $-T\Delta S^o$ | $K_a$ | $\Delta H^o$ | $-T\Delta S^o$ |
| G1 | 5a | $(4.5 \pm 0.2) \times 10^3$ | $-35 \pm 4$ | $14 \pm 4$ | $(2.5 \pm 0.2) \times 10^2$ | $-10.3 \pm 0.1$ | $-3.3 \pm 0.2$ |
| | 5b | $(4.4 \pm 0.1) \times 10^3$ | $-18 \pm 1$ | $-2 \pm 1$ | $(5.0 \pm 0.3) \times 10^2$ | $-15 \pm 2$ | $-1 \pm 2$ |
| G2 | 5a | $(8.4 \pm 0.1) \times 10^4$ | $-42.4 \pm 0.5$ | $14.3 \pm 0.5$ | $(2.4 \pm 0.4) \times 10^3$ | $-22 \pm 2$ | $2 \pm 2$ |
| | 5b | $(3.7 \pm 0.1) \times 10^5$ | $-36 \pm 1$ | $4.2 \pm 0.9$ | $(1.5 \pm 0.2) \times 10^4$ | $-30 \pm 3$ | $7 \pm 3$ |
| G3 | 5a | $(2.0 \pm 0.2) \times 10^4$ | $-29 \pm 2$ | $5 \pm 2$ | $(5.8 \pm 0.3) \times 10^2$ | $-20 \pm 2$ | $5 \pm 2$ |
| | 5b | $(9.4 \pm 0.3) \times 10^4$ | $-28 \pm 2$ | $-1 \pm 2$ | $(1.7 \pm 0.1) \times 10^3$ | $-19 \pm 1$ | $1 \pm 1$ |

[a]The data were averaged from three independent titrations.

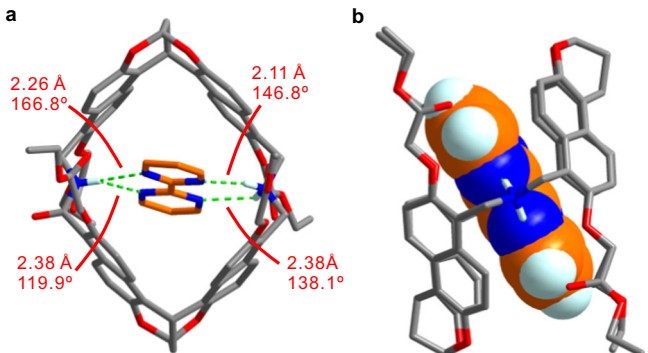

**Fig. 4 X-ray single crystal structure of G4@4b • H$^+$. a** The over view of **G4@4b • H$^+$** is presented as a stick model, showing four hydrogen bonds between guest and host. **b** Host and guest are shown with stick model and space-filling model in the side view of **G4@4b • H$^+$**, respectively. **G4** is fully encapsulated in the cavity of **4b•H$^+$**. Most of hydrogen atoms and PF$_6^-$ counter ions are omitted for viewing clarity. Color codes: carbon, gray or orange; oxygen, red; nitrogen, blue.

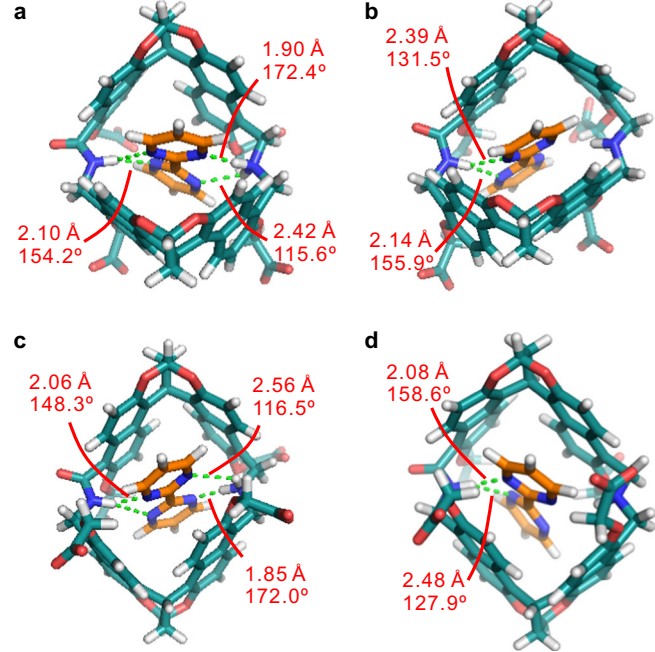

**Fig. 5 Binding modes between 5 and G4.** Energy-minimized structures of (**a**) **G4@5a • H$^+$**, (**b**) **G4@5a**, (**c**) **G4@5b • H$^+$** and (**d**) **G4@5b** calculated by DFT calculations (wB97XD/6–311 + G(d) and 6–311 G(d)) with the PCM solution model in water at 298 K. Color codes: carbon, celeste or orange; oxygen, red; nitrogen, blue.

To further reveal the binding mechanism in water at both pH values, DFT calculations on the complexes of **G4** with **5a** and **5b** in water were performed with the host to be protonated and deprotonated. As shown in Fig. 5, the binding mode and the geometry for the complex of **5b** and **G4** are similar to those of the crystal structure of the complex of **4b** and **G4** at the protonated state. This supports our reasoning that the binding mode in an *endo*-functionalized cavity should not be significantly affected by solvents. Similar binding mode was observed for the complex of **G4** and **5a**. At the deprotonated state, the guest only forms hydrogen bonds with the amide protons of the hosts. Thus, the binding affinity is significantly weakened at pH 12.

From the above analysis, protonation and deprotonation of the secondary amine of the naphthotubes drastically change the cavity feature and thus the direct noncovalent interactions between the hosts and the guests. This would then change the binding preferences of the hosts at the two pH values. For the complexes between **G5** and nonprotonated naphthotubes **5**, the binding is mainly driven by the hydrophobic effect and cation–π interactions. Nevertheless, tetramethylammonium shows very weak binding to **5** (Supplementary Fig. 88), which may be due to unmatched shape and size. In addition, the weak binding between **G5** and protonated naphthotubes **5** should result from the charge repulsion between the charged cavity and charged guests.

**Switchable Bifunctional molecular recognition**. To achieve switchable bifunctional molecular recognition, it needs two guests which are the best binders to the naphthotubes at the protonation

and deprotonation states, respectively. In addition, the two guests should also have drastically different binding affinities at the two states of the hosts to enable clear-cut differentiation. Naphthotube **5b** is usually a better binder to these guests than **5a**. Therefore, **5b** is selected as the host for the demonstration. Guests **G4** and **G5** have the largest differences in the association constants at the two states of **5b** and are thus selected as the guests.

First, the switchable binding was demonstrated in a binary system with **5b** and **G4/G5**. **5b** shows very high binding affinity to **G4** at pH 7.4 ($K_a = 1.7 \times 10^4$ M$^{-1}$), but binds very weakly to **G4** at pH 12 ($K_a = 90$ M$^{-1}$). In contrast, **5b** binds **G5** strongly ($K_a = 5.9 \times 10^3$ M$^{-1}$) at pH 12 and weakly ($K_a = 20$ M$^{-1}$) at pH 7.4. The large differences in the binding affinities at the two states would allow an on-off/off-on switch in the binary mixtures. Indeed, when increasing the pH values of the solution containing **5b** and **G4** at pH 7.4 through gradually adding 8 equivalents of NaOH, the proton signals of **G4** shift downfield to the position of free **G4** (Supplementary Fig. 89). The $^1$H NMR spectra is the superposition of free **G4** and free but deprotonated **5b** at pD 12. This suggests that **G4** is released from the cavity of **5b** when the

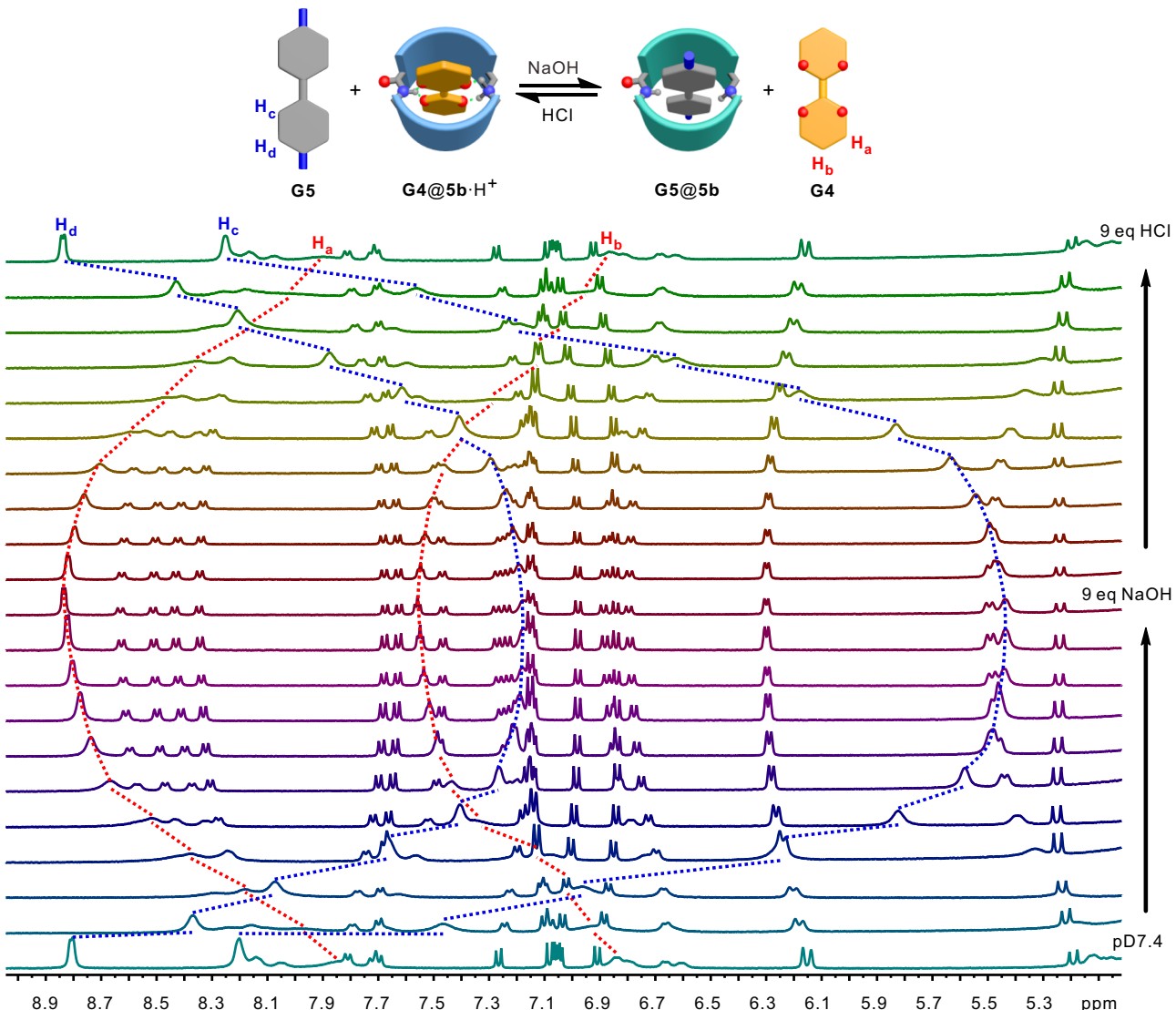

**Fig. 6 Switchable bifunctional molecular recognition.** Cartoon representation of switchable bifunctional molecular recognition and partial $^1$H NMR spectra (500 MHz, $D_2O$) of the equimolar mixture of **5b**, **G4** and **G5** at pD 7.4 after gradually adding 9 equivalents of NaOH and then 9 equivalents of HCl.

cavity is deprotonated after adding NaOH. The $^1$H NMR spectra can be restored after adding 8 equivalents of HCl. This indicates that **G4** can be up-taken again into the cavity of **5b** after protonation. The two processes are reversible. Similar phenomena are observed for the binary mixture of **5b** and **G5** (Supplementary Fig. 90). Differently, **G5** was taken up into the cavity of **5b** at the deprotonation state and kicked out from the cavity at the protonation state.

Finally, switchable bifunctional molecular recognition was demonstrated with the ternary mixture of **5b**, **G4** and **G5**. The solution was prepared in $D_2O$ and the initial pD is determined to be ca. 7.4. The $^1$H NMR spectra (Fig. 6) is the superposition of the spectra of free **G5** and the complex of **5b•H$^+$** and **G4** (Supplementary Fig. 91). After gradually adding 9 equivalents of NaOH, the $^1$H NMR spectra undergo large changes, and the spectrum is the superposition of the spectra of free **G4** and the complex of **5b** and **G5**. This suggests guest **G4** in the cavity of **5b•H$^+$** is kicked out and swapped with **G5** after converting **5b•H$^+$** to **5b** through deprotonation. This shows **5b** is able to perform switchable bifunctional molecular recognition in the ternary mixture of **5b**, **G4** and **G5**. Moreover, the system can be restored to its original state through adding 9 equivalents of HCl to the

above solution (Fig. 6), showing the good reversibility of the system.

In summary, we reported a pair of biomimetic macrocyclic hosts with a pH-responsive *endo*-functionalized cavity. These naphthotubes possess two inward-directing functional groups, amide and secondary amine, in their cavities. The secondary amine group can be protonated at pH 7.4 and deprotonated at pH 12. Thus, the cavity features of the naphthotubes can be switched between the two states. Therefore, the naphthotubes show drastically different guest-binding preferences at the two pH values. Further research reveals that the noncovalent interactions between the hosts and the guests are different at the two host states, leading to different binding preference and selectivity. The guest-binding behavior of the naphthotubes can be simply tuned by adjusting the pH value in the solution. In a ternary mixture of one host with two guests, the guest molecules can be selectively up-taken into the cavity of the host by changing the pH value and thus the cavity states. These biomimetic macrocyclic hosts with selective and switchable recognition properties at different pH values are interesting for water-soluble synthetic hosts. The binding affinity may be too small for noncovalent bioconjugation[25], but these hosts may still find applications in

the fields of self-assembly, water-soluble molecular machines, environmental sciences and biomedical sciences.

## Methods

**General**. All the reagents and guest molecules involved in this research were commercially available and used without further purification unless otherwise noted. $^{1}H$, $^{13}C$ NMR, $^{1}H$–$^{1}H$ COSY and $^{1}H$–$^{1}H$ ROESY NMR spectra were recorded on Bruker Avance-400 (500) spectrometers. Electrospray-ionization time-of-flight high-resolution mass spectrometry (ESI-TOF-HRMS) experiments were conducted on an applied Q EXACTIVE mass spectrometry system. Fluorescence spectra were obtained on a Shimadzu RF-5301pc spectrometer.

**Synthesis and characterization**. Synthesis and the corresponding characterization data are provided in the Supplementary Information.

**Determination of the association constants**. To determine the association constants, NMR titrations and ITC titrations were performed in phosphate buffer (50 mM, pH = 7.4 or 12) at 298 K. NMR titrations were carried out by adding guest to the solution of host with a fixed concentration. Non-linear curve-fitting was then performed on the plots of $\delta_{obs}$ of host as a function of [G] to obtain the association constants ($K_a$). In a typical ITC titration experiment, a 1.4338 mL solution of host was placed in the sample cell, and 292 µL of a solution of guest was in the injection syringe. Heats of dilution, measured by titration of guest into the sample cell with blank solvent, were subtracted from each data set. All solutions were degassed prior to titration. The data were analyzed using the instrumental internal software package and fitted by "one set of binding sites" model to give the association constants ($K_a$). Non-linear fitting data are shown in Supplementary Figs. 12–21, 54–73 and 88. ITC titration data are shown in Supplementary Figs. 74–87.

## Data availability

The X-ray crystallographic coordinates for structures reported in this study have been deposited at the Cambridge Crystallographic Data Centre (CCDC), under deposition numbers 2106751 (**4a**), 2106752 (**4b**) and 2106753 (**G4@4b**). These data can be obtained free of charge from the Cambridge Crystallographic Data Centre via www.ccdc.cam.ac.uk/data_request/cif. All other data supporting the findings of this study are available within the Article and its Supplementary Information and/or from the corresponding author upon request.

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

## Acknowledgements

This research was financially supported by the National Natural Science Foundation of China (No. 22125105, W.J.), the Shenzhen Science and Technology Innovation Committee (JCYJ20180504165810828, W.J.), Shenzhen "Pengcheng Scholar", Guangdong Provincial Key Laboratory of Catalysis (No. 2020B121201002, W.J.), and Guangdong High-Level Personnel of Special Support Program (2019TX05C157, W.J.). We are grateful to Dr. Xiaoyong Chang for the help on X-ray crystallography, the technical support from SUSTech-CRF and the Center for Computational Science and Engineering of SUSTech.

## Author contributions

W.J. conceived and designed the experiments. X.W. carried out the experimental work. M.Q. performed the DFT calculations. H.Y., X.-Y.P. and H.K. solved the crystal structure. W.J. and X.W. analyzed the data and wrote the manuscript, and all the authors commented on it.

## Competing interests

The authors declare no competing interests.
