## [Peer Review File · Nature Communications]

Switchable Bifunctional Molecular Recognition in Water Using a pH-Responsive Endo-Functionalized CavityREVIEWER COMMENTS

Reviewer #1 (Remarks to the Author):

The Authors reported a beautiful work of biomimetic macrocyclic hosts with a pH-responsive endo-functionalized cavity by using his new naphthotubes, which possess two inward-directiong functional groups-amide and secondary amine. The secondary amine group can be protonated at pH 7.4 and deprotonated at pH 12. So the cavity features of the naphthotubes can be switched between the two states. The author confirmed that the naphthotubes show different guest-binding preferences at the two pH values, which related to the noncovalent interactions in the complexes are different. They also showed the switchable recognition properties at different pH values, which may be applied in the fields of chemical biology, molecular machines and biomedical sciences. So I think this will be suitable to be published in Nature Communications for its broad readers.

However, there are several minor questions should be answered before published:

1. Fig. 2a, I don't think this arrow is suitable here.
2. Cartoon representation for switchable bifunctional molecular recognition is different between Fig. 2b and Fig. 6. And the author just did one experiment for switchable bifunctional molecular recognition.

Reviewer #2 (Remarks to the Author):

The manuscript entitled "Switchable Bifunctional Molecular Recognition in Water Using a pH-Responsive Endo-Functionalized Cavity" that Jiang and co-workers submitted to be considered for publication in Nat. Commun. Describes the synthesis of two nafto-tubes isomeric receptors equipped with a polar cavity. The net charge of the cavity can be modified from neutral to positive and vice-versa owing to a protonation/deprotonation process exerted by a change in the pH of the solution. The receptor tubes show different binding properties for a series of guests in the tow different states: neutral or protonated. This difference in binding properties is maximized in the case of the pair of guest G4 (neutral) and G5 (positively charged). The authors performed pairwise competitive binding experiments between the two guests at different pH demonstrating the selectivity and reversibility of the binding processes. I found the reported work very interesting. I am convinced that it will be of interested to the wide readership of the journal. For this reason, I recommend the acceptance of the work for publication after the authors have addressed the minor issues listed below.

1. G5 is particular, it binds the receptors not making use of the polar groups of the cavity. Its binding is mainly driven by the hydrophobic effect. This should be mentioned in the text. Are other alkylammonium molecules bound in the neutral cavity? i.e. TMA, TBA or alike?
2. In the abstract, the authors suggested/mentioned the potential use of the developed system for noncovalent molecular/bio-conjugation. However, the reported binding constants for the formed complexes, either in the neutral or positively charged state of the receptor, are to small for this endeavor. The literature covering the topic is correctly cited. However, the limitation of the system to be use for non-covalent conjugation is not mentioned. I recommend removing the first sentence of the abstract and commenting the above mentioned limitation in the text.

Reviewer #3 (Remarks to the Author):

The work by Jian reports a novel host able to change its binding capabilities as function of pH. In particular, shifting among pHs protonation of one amine in the internal cavity of the host has important effects in the binding properties. These interesting results have been obtained screening different hosts and guests.

In my opinion the paper is interesting and inspiring. Moreover, I think that there is sufficient experimental material to support the conclusions.

For these reasons I think that the paper can be accepted as it is.

Reviewer #4 (Remarks to the Author):

The manuscript reports the preparation of a macrocyclic receptor with an inward-facing secondary amine group which can be protonated and deprotonated. Both forms can be accessed and remain soluble in water through changing the pH. The different protonation states of the host show different guest binding preferences as might be expected which is used to realize pH controlled switching between two different guests in solution. The new receptors are closely related to previous hosts reported by the authors (described as naphthotubes) but the pH switching properties are new in this work. The design features of the host which allow the switching to occur are also discussed as well as the roles of intermolecular interactions in controlling the different binding affinities of the two states.

The solution behaviour and binding studies have generally been well characterised and all data presented support the authors claims. The introduction does oversell the work somewhat and ignores many existing examples of pH controlled binding and stimuli-responsive hosts (eg DOIs 10.1039/C9OB00398C, 10.1021/jacs.5b05960 and 10.1038/s41467-019-09928-x to give just a few). There is also too much use of buzz words like biomimetic and molecular machines which are not directly relevant to the present work.

Other minor comments:

Errors in Table 2 should be given to 1 significant figure.
13C NMR peaks should be reported to 1 decimal place

Comments on X-ray crystallography:

The X-ray crystal structure data is sufficient to support the authors claims but some aspects of the refinements should be improved and the structure of G4@4b re-examined before publication as described below.

All cif files:

Hydrogen atoms attached to oxygen and nitrogen should be located from the electron density maps (with DFIX restraints where necessary) rather than positioned geometrically where possible. Explanations should be given where this was not possible.

Details of hydrogen bonds should be entered into the .cifs (through use of HTAB during refinement)
An explanation should be given for all B level alerts in the cif files. Preferably by filling out the VRF forms. Some of the B level alerts can easily be fixed (see below) and should be when feasible.

Structure specific problems:

4a

- H atoms on C54 and C66 may be positioned incorrectly (also see comment below).
- Bond between C53 and C54 is too short. Likely due to disorder of C54 which should be modelled if possible.
- Data has been cut-off at 0.9 Å but mean I/sigma is still >5. Ideally data should be integrated a bit further.
- Atoms should be sorted in a logical order within the .ins file.

G4@4b

- Platon suggests possible space group of C2/c. Structure should be carefully checked for additional symmetry to see if this is genuine or only pseudo-symmetry and final choice of space group justified.
- There are a lot of very poor thermal parameters which the authors have tried to fix with strong ISOR restraints instead of dealing with the underlying problem which is most likely the incorrect choice of space group or possibly twinning (the data does show some signs of twinning). The structure is not publishable in the current form until these issues are addressed.
- Data has been cut-off at 0.9 Å but mean I/sigma is still >5. Ideally data should be integrated a bit further.
- Atoms should be named and sorted in a logical order within the .ins file.

Overall this is a nice piece of work which will be of general interest to the supramolecular chemistry

community once suitable revisions have been made.

Point-by-Point Response

REVIEWER COMMENTS

Reviewer #1 (Remarks to the Author):

The Authors reported a beautiful work of biomimetic macrocyclic hosts with a pH-responsive endo-functionalized cavity by using his new naphthotubes, which possess two inward-directional functional groups-amide and secondary amine. The secondary amine group can be protonated at pH 7.4 and deprotonated at pH 12. So the cavity features of the naphthotubes can be switched between the two states. The author confirmed that the naphthotubes show different guest-binding preferences at the two pH values, which related to the noncovalent interactions in the complexes are different. They also showed the switchable recognition properties at different pH values, which may be applied in the fields of chemical biology, molecular machines and biomedical sciences. So I think this will be suitable to be published in Nature Communications for its broad readers.

Response: Thank you very much for the kind recommendation and helpful suggestions!

However, there are several minor questions should be answered before published:

1. Fig. 2a, I don't think this arrow is suitable here.

Response: This has been changed accordingly. The arrow has been removed.

2. Cartoon representation for switchable bifunctional molecular recognition is different between Fig. 2b and Fig. 6. And the author just did one experiment for switchable bifunctional molecular recognition.

Response: Thank you for bringing this into our attention! The cartoon representations in Fig. 2c and Fig. 6 have been changed accordingly to be consistent. In fact, we have done a series of experiments to illustrate the switchable bifunctional molecular recognition. Only two guests were selected for the demonstration because these two guests have the largest contrast in the binding affinities to the two states of the hosts.

Reviewer #2 (Remarks to the Author):

The manuscript entitled "Switchable Bifunctional Molecular Recognition in Water Using a pH-Responsive Endo-Functionalized Cavity" that Jiang and co-workers submitted to be considered for publication in Nat. Commun. Describes the synthesis of

two naphtho-tubes isomeric receptors equipped with a polar cavity. The net charge of the cavity can be modified from neutral to positive and vice-versa owing to a protonation/deprotonation process exerted by a change in the pH of the solution. The receptor tubes show different binding properties for a series of guests in the two different states: neutral or protonated. This difference in binding properties is maximized in the case of the pair of guest G4 (neutral) and G5 (positively charged). The authors performed pairwise competitive binding experiments between the two guests at different pH demonstrating the selectivity and reversibility of the binding processes. I found the reported work very interesting. I am convinced that it will be of interest to the wide readership of the journal. For this reason, I recommend the acceptance of the work for publication after the authors have addressed the minor issues listed below.

Response: Thank you very much for the kind recommendation and valuable suggestions! We have changed the manuscript accordingly.

1. G5 is particular, it binds the receptors not making use of the polar groups of the cavity. Its binding is mainly driven by the hydrophobic effect. This should be mentioned in the text. Are other alkylammonium molecules bound in the neutral cavity? i.e. TMA, TBA or alike?

Response: Thank you very much for bringing this into our attention! The following paragraph has been added in the main text: “For the complexes between G5 and nonprotonated naphthotubes 5, the binding is mainly driven by the hydrophobic effect and cation- π interactions. Nevertheless, tetramethylammonium shows very weak binding to 5, which may be due to unmatched shape and size. In addition, the weak binding between G5 and protonated naphthotubes 5 should result from the charge repulsion between the charged cavity and charged guests.”

We determined the binding constants between TMA and 5a/5b through NMR titrations. The binding constants are quite small (see below). This may be due to unmatched shape and size, as these guests are spherical and the naphthotubes are tubular and narrow in

size. This was also included in the maintext and supplementary materials.

2. In the abstract, the authors suggested/mentioned the potential use of the developed system for noncovalent molecular/bio-conjugation. However, the reported binding constants for the formed complexes, either in the neutral or positively charged state of the receptor, are too small for this endeavor. The literature covering the topic is correctly cited. However, the limitation of the system to be used for non-covalent conjugation is not mentioned. I recommend removing the first sentence of the abstract and commenting on the above-mentioned limitation in the text.

Response: Thank you for this suggestion!

The first sentence of the abstract has been rephrased according to your suggestions. In addition, we also added the comment on the limitation of the system for noncovalent conjugation in the conclusion part.

Reviewer #3 (Remarks to the Author):

The work by Jian reports a novel host able to change its binding capabilities as a function of pH. In particular, shifting among pHs, protonation of one amine in the internal cavity of the host has important effects on the binding properties. These interesting results have been obtained by screening different hosts and guests.

In my opinion, the paper is interesting and inspiring. Moreover, I think that there is sufficient experimental material to support the conclusions.

For these reasons, I think that the paper can be accepted as it is.

Response: Thank you for the kind recommendation!

Reviewer #4 (Remarks to the Author):

The manuscript reports the preparation of a macrocyclic receptor with an inward-facing secondary amine group which can be protonated and deprotonated. Both forms can be accessed and remain soluble in water through changing the pH. The different protonation states of the host show different guest binding preferences as might be expected which is used to realize pH controlled switching between two different guests in solution. The new receptors are closely related to previous hosts reported by the authors (described as naphthotubes) but the pH switching properties are new in this work. The design features of the host which allow the switching to occur are also discussed as well as the roles of intermolecular interactions in controlling the different binding affinities of the two states.

Response: Thank you very much for the recommendation and very helpful suggestions and comments!

The solution behaviour and binding studies have generally been well characterised and all data presented support the authors claims. The introduction does oversell the work somewhat and ignores many existing examples of pH controlled binding and stimuli-responsive hosts (eg DOIs 10.1039/C9OB00398C, 10.1021/jacs.5b05960 and 10.1038/s41467-019-09928-x to give just a few). There is also too much use of buzz words like biomimetic and molecular machines which are not directly relevant to the present work.

Response: Thank you for these suggestions!

In fact, we have cited many review articles on known stimuli-responsive hosts (please see refs. 31-35). There are many research articles on stimuli-responsive hosts, which are not the same as the hosts reported here. We think these review articles would be representative and cover most of these research articles. Nevertheless, the three papers suggested are still cited (please see refs. 36-38 in the revised manuscript). In addition, the corresponding sentence has been changed slightly. Some of the words "biomimetic" have been removed.

Other minor comments:

Errors in Table 2 should be given to 1 significant figure.

¹³C NMR peaks should be reported to 1 decimal place

Response: Thank you! This has been changed accordingly.

Comments on X-ray crystallography:

The X-ray crystal structure data is sufficient to support the authors claims but some aspects of the refinements should be improved and the structure of G4@4b re-examined before publication as described below.

All cif files:

Hydrogen atoms attached to oxygen and nitrogen should be located from the electron density maps (with DFIX restraints where necessary) rather than positioned geometrically where possible. Explanations should be given where this was not possible. Details of hydrogen bonds should be entered into the .cif files (through use of HTAB during refinement)

An explanation should be given for all B level alerts in the cif files. Preferably by filling out the VRF forms. Some of the B level alerts can easily be fixed (see below) and should be when feasible.

Structure specific problems:

4a

- H atoms on C54 and C66 may be positioned incorrectly (also see comment below).
- Bond between C53 and C54 is too short. Likely due to disorder of C54 which should be modelled if possible.
- Data has been cut-off at 0.9 Å but mean I/sigma is still >5. Ideally data should be integrated a bit further.
- Atoms should be sorted in a logical order within the .ins file.

G4@4b

- Platon suggests possible space group of C2/c. Structure should carefully be checked for additional symmetry to see if this is genuine or only pseudo-symmetry and final choice of space group justified.
- There are a lot of very poor thermal parameters which the authors have tried to fix with strong ISOR restraints instead of dealing with the underlying problem which is most likely the incorrect choice of space group or possibly twinning (the data does show some signs of twinning). The structure is not publishable in the current form until these issues are addressed.
- Data has been cut-off at 0.9 Å but mean I/sigma is still >5. Ideally data should be integrated a bit further.
- Atoms should be named and sorted in a logical order within the .ins file.

Overall this is a nice piece of work which will be of general interest to the supramolecular chemistry community once suitable revisions have been made.

Response: Thank you very much for the detailed suggestions on the crystal structures! All the crystal structures have been refined accordingly. New structures have been uploaded to CCDC and together with this revised manuscript.

REVIEWERS' COMMENTS

Reviewer #2 (Remarks to the Author):

I am totally satisfied with the changes made by the authors in the revised version. I support the acceptance of the ms "as is" for publication.

Reviewer #4 (Remarks to the Author):

The authors have satisfactorily responded to my comments and those of the other referees and I'm now happy to see this excellent work published in Nat. Commun..

One small comment - the values in Table 1 and 2 should be rounded such that the error is in the last significant digit. i.e 6 ± 1 rather than 5.8 ± 1 .

Point-by-Point Response

REVIEWER COMMENTS

Reviewer #2 (Remarks to the Author):

I am totally satisfied with the changes made by the authors in the revised version. I support the acceptance of the ms "as is" for publication.

Response: Thank you very much for the kind recommendation!

Reviewer #4 (Remarks to the Author):

The authors have satisfactorily responded to my comments and those of the other referees and I'm now happy to see this excellent work published in Nat. Commun..

Response: Thank you very much for the kind recommendation!

One small comment - the values in Table 1 and 2 should be rounded such that the error is in the last significant digit. i.e 6 ± 1 rather than 5.8 ± 1 .

Response: Thank you for bringing this into our attention! This has been changed accordingly.